

# Assessing terrestrial carbon fluxes and stocks in South America and its major biomes using CMIP6 Earth System Models

Marcos B. Sanches[1], Manoel Cardoso[1], Celso von Randow[1], Chris Jones[2], Mathew Williams[3]

[1] Impacts, Adaptation and Vulnerabilities Division, National Institute for Space Research, São José dos Campos, Brazil.
5 [2] Met Office Hadley Centre, Exeter, UK.
[3] University of Edinburgh, Edinburgh, UK

*Correspondence to*: Marcos B. Sanches (marcos.barbosa.sanches@gmail.com)

**Abstract.** South America plays a significant role in the global carbon cycle, with its ecosystems storing substantial amounts of carbon in vegetation and soils. This study analyses the behaviour of components of the carbon cycle and stocks in the 10 whole continent and two of its major biomes, the Amazon and the Savannas, using a set of 18 Earth System Models (ESMs) from the Coupled Model Intercomparison Project Phase 6 (CMIP6). We discuss the variability of the model simulations throughout the 20th and first 20 years of the 21st centuries. Results show that South America accounts for 25-30% of the global Gross Primary Productivity (GPP), 21-28% of the global Net Primary Productivity (NPP), 17-50% of the global Net Ecosystem Productivity (NEP), and 15-30% of the global Net Biome Productivity (NBP), and also contributes significantly 15 to global autotrophic (Ra) and heterotrophic (Rh) respiration. The temporal evolution of NBP in South America indicates a combination of the values estimated for the Amazon and the Savannas, with most models showing a small decreasing trend in the 20th century, likely dominated by emissions from land use change, and shifting to positive values after 1990, likely driven by an increasing productivity in response of atmospheric $CO_2$ fertilization. Comparing the Amazon and Savannas, apart from the magnitude of the fluxes, we see similarities in both ecosystems responses when widespread dry years occur, 20 with higher NBP and GPP in wet years and higher Rh and disturbances in dry years. These results highlight the vulnerability of South America to climate change, with the potential for parts of the continent to shift from carbon sinks to sources under widespread droughts.

## 1 Introduction

Carbon is a critical element in the Earth system. Together with other greenhouse gases such as water vapor and nitrous 25 oxide, atmospheric carbon, in the form of carbon dioxide ($CO_2$) and methane ($CH_4$), regulates the Earth's climate by trapping heat in the atmosphere. As the main element of organic compounds, it is often referred to as the "element of life", being stored in different reservoirs, such as the ocean, the atmosphere, soils and terrestrial biomass and transported between these reservoirs through the global carbon cycle. Without external disturbances, this cycle maintains a stable concentration in the atmosphere, oceans and land (IPCC, 2023). However, changes in human activity since the Industrial Era have resulted in 30 increasing emission rates of fossil fuels, which, together with the conversion of natural landscapes to agricultural land, has



emitted about 35 Pg of $CO_2$ annually into the atmosphere (Friedlingstein et al, 2022; IPCC, 2023). Consequently, among several developments, there has been a significant increase in atmospheric $CO_2$, rising from 280 ppm in pre-industrial times to almost 420 ppm in 2023.

This increase in atmospheric $CO_2$ has affected the global climate in several ways by influencing multiple processes in the
biosphere, such as photosynthesis and plant respiration, which are influential on precipitation patterns and global temperature (Jung et al, 2017).

Currently, terrestrial ecosystems sequester approximately one-third of anthropogenic greenhouse gas emissions every year (Le Quéré et al, 2018), which are mostly absorbed by tropical forests, such as those in South America, where the largest carbon sinks are located. However, the increased frequency of hot spells and the prolongation of regional droughts may
reduce this efficiency to the point where these biomes shift from sinks to emitters. In 2024, extensive and persistent areas of burning in the Brazilian Amazon, Cerrado and Pantanal biomes in Brazil generated a continuous source of carbon emissions with a record that broke the record of the last 22 years of monitoring carried out by CAMS (acronym in English for Copernicus Atmosphere Monitoring Service) - approximately 0,076 PgC, 47% of which was in the month of September alone. The high temperatures that South America experienced this year, the long-term drought that contributed to low soil
moisture, in addition to other climatological factors probably contributed to the significant increase in the incidence of fires, smoke and impacts on air quality. (Copernicus, 2024; Axios, 2024).

Earth system models (ESMs) simulate the Earth's climate system iteratively by coupling physical general circulation models of the atmosphere, ocean, land and cryosphere with biogeochemical and biophysical cycles (Jones, 2020). The results contribute to improving knowledge in the various areas that make up the Earth system, such as understanding the responses
of the carbon cycle to climate variability in the face of climate change (Friedlingstein et al., 2019). In particular, analyses of ESM results for South America can provide relevant information to projects such as the Regional Carbon Cycle Assessment and Processes (RECCAP), and the Climate Science for Service Partnership Brazil (CSSP Brazil). Even though there are large inaccuracies among models in the representation of both the terrestrial carbon cycle and land use (Le Quéré et al., 2018), in capturing the interactions between the terrestrial biosphere and the atmosphere and the relationships between water
and nutrients (Roxburgh et al., 2004), in the balance between the strong influence of $CO_2$ fertilization and the negative effects of rising temperatures and drought to simulate tree mortality, these models are crucial tools for projecting the evolution of the global terrestrial carbon cycle both for a better understanding of past patterns of various properties of tropical forests and for projections in different future scenarios, which assists in society's political decisions (Koch et al., 2021).

For a consistent comparison between different ESMs, the Coupled Model Intercomparison Project (CMIP; Meehl et al., 2000) was created at the end of the 20th century, through the use of uniform initialization and output structures, with the aim of understanding past, present and future climate. The most recent phase, CMIP6 (Eyring et al., 2016a), provides a large set of model simulations.





This paper develops several analyses on the behaviour of fluxes and stocks within the carbon cycle, for all of South America
and two of its biomes, the Amazon and the Savannah areas. We discuss the quality with which the CMIP6 models represent
the components of the carbon balance compared to previously published and estimated results. We address the susceptibility
of the continent and its biomes to become a potential carbon emitter in the face of the increasing frequency of dry years,
verifying which variables best respond to the variability of this process within a carbon cycle affected by climate change.

## 2. Methods

### 2.1 Terrestrial carbon fluxes and stocks

The scope of fluxes and stocks in the terrestrial carbon cycle can be seen in Fig. 1. Plant photosynthesis causes a net
absorption of $CO_2$ from the atmosphere by terrestrial ecosystems of about 120 PgC yr$^{-1}$. This process is expressed in the
form of a flux and is known as gross primary production (GPP). Approximately 50% of the carbon from this photosynthetic
process is released directly by autotrophic respiration (Ra) (Gifford, 2003). The part fixed in plants is used for the growth
and development of leaves, stems and roots. This process in which plants convert the amount of photosynthetic matter into
biomass is called net primary production (NPP) (Malhi et al., 2011). After the death of a plant or parts of it, residues are
produced. Heterotrophic respiration (Rh) is essentially the process of oxidation of organic matter in the litter, releasing this
carbon flux into the atmosphere (Chapin et al., 2006; Bond-Lamberty, 2017). The net flux of $CO_2$ to or from a forest
ecosystem is called Net Ecosystem Production (NEP) or Net Ecosystem Exchange (NEE). NEE quantifies the efficiency of
an ecosystem in storing terrestrial carbon. Keenan et al., 2016, proposed that terrestrial ecosystems have sequestered around
30% of annual anthropogenic emissions. Net Biome Production (NBP) is the flux that measures the change in carbon stocks
after accounting for losses due to natural or anthropogenic disturbances such as fire, deforestation, water fluxes, severe
downbursts, and timber products. NBP provides an estimate of the amount of carbon stored in a biome over a given period.
Vegetation carbon (cVEg) or biomass represents the carbon reservoir in the leaf, stem, root, as well as other plant
components such as fruits. Carbon stored in the soil (cSoil) is found in organic (plant and animal residues, microbes and
microbial by-products) and inorganic (carbon minerals produced by the weathering of the original material, or reaction of
soil minerals with atmospheric carbon dioxide) forms.





**Figure 1.** The green flux represents carbon uptake by vegetation. The red fluxes indicate carbon release through plant respiration (Ra), soil respiration (Rh), and disturbances, whether natural or anthropogenic, such as fire. The sum of respiratory fluxes constitutes the total ecosystem respiration (Reco). The balance between gross primary production (GPP) and plant respiration (Ra) defines the net primary production (NPP). The difference between NPP and Rh describes the net ecosystem production (NEP). Finally, the balance between NEP and all carbon loss fluxes determines the net biome production (NBP).




## 2.2 Earth System Models

In this paper, we use a set of 18 ESMs, all of which followed the CMIP6 simulation protocols. The details can be seen in
Table 1, with the characteristics of the models, their atmospheric and terrestrial components, and their main references. The
models were selected according to their availability in the Earth System Grid Federation (ESGF) repositories for the
variables considered. We present historical simulations between the 20th and early 21st centuries up to 2020. Following the
conventional use of these sets, we highlight the multi-model ensemble, but we also analyse the individual performance of the
models in simulating carbon fluxes and stocks at different regional scales, aiming to relate these results to the scope of the
processes represented in each model.

Jones et al. (2023) performed a detailed assessment of regional carbon budgets to analyse the terrestrial carbon cycle in
CMIP6 models. The results indicated that multi-model ensembles perform well across most regions and variables, both for
carbon fluxes and stocks, although individual models have different strengths and limitations. In addition, it is possible to
have a more complete understanding from the combination of different approaches (Kondo et al, 2020 and Ciais et al, 2020),
and a more robust investigation of which processes are driving carbon fluxes and how they vary in space and time. Multi-
model ensembles also allow for more precise identification of areas of uncertainty or where more research is needed, based
on comparative cross-validation of results from different models (Mastrandrea et al., 2011, Hewitt et al., 2016 and Booth et
al., 2017).

ESMs in the set that have explicit modelling of fire, its occurrence and effects on vegetation are: CESM2, CESM2-
WACCM, CNRM-ESM2-1, GFDL-ESM4, MIROC-ES2L, MPI-ESM1-2-LR, NorESM2-LM, NorESM2-MM. There is also
a subset of ESMs that have an improvement in the interaction between the Carbon and Nitrogen cycles (ACCESS-ESM1-5,
CESM2, CESM2-WACCM, CMCC-CM2-SR5, EC-Earth3-Veg, MIROC-ES2L, MPI-ESM1-2-LR, NorESM2-LM,
NorESM2-MM, TaiESM1, UKESM1-0-LL). In version 5 of CMIP, an overestimation in photosynthesis was detected due to
the difficulty of surface models in reproducing, in biogeochemical fields, the limitations that the nitrogen cycle lay on the
absorption of $CO_2$ (Anav et al., (2013a)). This interaction possibly alters the way in which carbon is absorbed and stored by
plants and also limits a possible overestimation of Net Primary Productivity (NPP) in a scenario of rising temperatures, in
addition to improving the spatial resolution in the estimation of carbon exchanges between the atmosphere and the terrestrial
biosphere (Mcguire et al., 1992).

An analysis from a model-data fusion system, CARDAMOM, provided a systemic benchmark for comparison with the
ESMs. CARDAMOM (Bloom et al., 2016) calibrates an intermediate complexity C cycle model, DALEC (Bloom and
Williams, 2015) according to gridded time series of climate and disturbance drivers, and observations of leaf area index time
series, woody biomass and soil C. Independent calibrations were generated for 1° pixels across South America consistent
with local observations. Thus, parameters are emergent from model and data, rather than set by plant functional types as in



the ESMs. CARDAMOM outputs were generated monthly from 2010-2019 consistent with data availability. More details on
the approach used can be found in (Smallman et al., 2021).

**Table 1.** CMIP6 models analyzed in this study.

| MODEL Country | ATMOSPHERE MODEL | LAND | | | |
|---|---|---|---|---|---|
| | | LAND MODEL | NUMBERS OF PLANT FUNCTIONAL TYPES (PFTs) | LAND USE CHANGE (L) FIRE MODEL (F) INTERACTIVE NITROGEN CYCLE (N) | MAIN REFERENCE |
| ACCESS-ESM1-5 Australia | UM7.3 | CABLE2.4, CASA-CNP | 13 | - - N | Law et al., (2017); Ziehn et al., (2017, 2019, 2020) |
| BCC-CSM2-MR China | BCC_AGCM3_MR | BCC-AVIM2 | 16 | L - - | Wu et al., (2019) |
| CanESM5 Canada | CanAM5 | CLASS3.6, CTEM1.2 | 13 | L - - | Swart et al., (2019a, 2019b) |
| CESM2 USA | CAM6 | CLM5 | 22 | L F N | Gettelman et al., 2019; Danabasoglu et al., (2019, 2020) |
| CESM2-WACCM USA | WACCM6 | CLM5 | 22 | L F N | Liu et al., 2019; Danabasoglu et al., (2019, 2020) |
| CMCC-CM2-SR5 Italy | CAM5 | CLM4.5 | 16 | - - N | Cherchi et al, 2019; Lovato and Peano (2020) |
| CNRM-ESM2-1 France | ARPEGE-Climate v6.3 SURFEXv8.0 | ISBA CTRIP | 16 | L F - | Séférian et al., (2018, 2019) |
| EC-Earth3-Veg Consortium - Europe | IFS36r4 HTESSEL | LPJ-GUESS | 12 | L F N | Döscher et al., (2022) |
| GFDL-ESM4 USA | AM4.1 | LM4.1 | 6 | L F - | Dunne et al., (2020) Krasting et al., (2018a, 2018b) |
| INM-CM4-8 Russia | integraded | integraded | -- | - - - | Volodin et al., (2018, 2019a) |
| INM-CM5-0 Russia | integraded | integraded | -- | - - - | Volodin et al., (2017a, 2017b, 2019b) |
| IPSL-CM6A-LR France | LMDZ6A | ORCHIDEEv2 | 15 | L - - | Boucher et al., (2018, 2020) |
| MIROC-ES2L Japan | MIROC-AGCM + SPRINTARS | VISIT-e & MATSIRO6 | 12 | L F N | Hajima et al., (2019, 2020a, 2020b) |
| MPI-ESM1-2-LR Germany | ECHAM6.3 | JSBACH3.2 | 13 | L F N | Mauritsen et al., (2019)/ Wieners et al., (2019a, 2019b) |
| NorESM2-LM Norway | CAM6 modified | CLM5 | 22 | L F N | Seland et al., (2019, 2020) |
| NorESM2-MM Norway | CAM6 modified | CLM5 | 22 | L F N | Seland et al., (2019, 2020) |
| TaiESM1 Taiwan, China | CAM5.3 modified | CLM4 modified | 16 | - - N | Lee et al., (2020); Lee and Liang (2020); Wang et al., 2021 |
| UKESM1-0-LL UK | Unified Model + UKCA | JULES-ES-1.0 | 13 | L - N | Tang et al., (2019); Sellar et al., (2019) |

**2.3 Study area**

South America has an estimated area of 17,819,100 km², its large north-south extension, associated with the heterogeneity of
the topography, favours a great variability of the climate (Garreaud et al., 2009; Espinoza et al., 2020). The ecosystems in
South America are privileged by their diversity and great influence on the global carbon cycle, as they store a significant
amount of carbon in their living biomass and soils, 140 PgC and 158 PgC respectively (Jones et al., 2023), which represents
more than 30% of the total found on the globe.

The Amazon biome (Fig. 2) has an estimated area of ~6.38 million km2, with relatively low climatic seasonality, annual
accumulated precipitation of >1,300 mm and average altitude of ≤1,000 m. There is a large concentration of biomass above





ground, with trees that can reach tens of meters (Eva et al., 2005; Castanho et al., 2013; Cardoso et al., 2017; Flores et al., 2024).

The savannas of South America (Fig. 2) have an estimated area of ~3.18 million km2, comprising a large part of the central area in Brazil, the west of the Northeast region, enclaves in Amapá, Roraima and Amazonas states, and outside Brazil we
find extracts in Venezuela, Colombia, southern Bolivia, northern Paraguay and Argentina. Precipitation varies between 600 mm and 2200 mm and its dry period can reach 6 months. Its vegetation cover is predominantly dense pastures, a cover of shrubs and trees that can range from sparse to closed areas, where the height of the trees varies between 12 and 15 meters (Maria et al, 2002; Schmidt et al., 2009; Bridgewater et al., 2004).


## BIOMES AREA

**Figure 2.** Amazonian biome area (blue) and South American Savannas biome areas (brown).

## 2.4 Selection of wet and dry years

Climate can have a strong influence on the carbon cycle. To better understand this, a biographical search was initially carried
out to determine the number of dry and wet years for different areas of South America. ESPINOZA et al., 2019, carried out a





survey based on observational data on the Amazon biome and presented the different years throughout the 20th and 21st centuries where wet and dry extremes occurred. When comparing these years with the meteorological conditions simulated by the models that make up this work, the result was mostly inconsistent. In other words, the physical and contour conditions of the models in different situations did not provide a match for whether it was a year of lack or excess of rain.

Therefore, an analytical criterion was proposed, based on the annual precipitation anomaly, counting the number of cells or grid points with negative and positive bias. We established a dry or wet year when two-thirds (2/3) of the chosen area had negative or positive anomalies, respectively. Marengo et al, 2008 and Papastefanou et al, 2022 observed that in situations of severe drought in the Amazon, the area affected by precipitation or water deficit covered more than 50% of the entire biome. This method is viable for this type of analysis because it is applicable and comprehensive for any spatial scale or fraction of

the study area. The climatology was obtained for each model between the years 1991-2020 as established by the WMO (World Meteorological Organization).

## 3. Results

### 3.1 South America annual fluxes and stocks

Figure 3 shows the total annual productivity fluxes computed over the entire South American continent, for each model.
Based on these models, GPP increased throughout the analysed period, with higher increase rates after 1960 (Fig. 3a). Initially, the average values are ~30 PgC yr$^{-1}$, with minimum values of 22 PgC yr$^{-1}$ and maximum values of 40 PgC yr$^{-1}$. Closer to the present, an average of ~34 PgC yr$^{-1}$ is observed, and extremes ranging from 25 PgC yr$^{-1}$ to 45 PgC yr$^{-1}$. It is also possible to note that there are two groups of results, one with values always above the average, and another with values close to or below the average among the models. For reference, values already proposed based on observations are 33.02 +/-
5.23 PgC yr$^{-1}$ for the years 2000-2019 (Jones et al., 2023), 33.66 PgC yr$^{-1}$ for 2000-2016 (Zhang et al., 2017), 36.08 PgC yr$^{-1}$ for 1979-2018 (FLUXCOM, Jung et al., 2020), and 33.21 PgC yr$^{-1}$ for the years 2010-2019, estimated by the structured data-model fusion tool "the CARbon DAta MODel fraMework", CARDAMOM (Bloom et al., 2016).

NPP in the region also presents a temporal pattern of growth, which intensifies after 1960 (Fig. 3b). Initially, the average value is ~13 PgC yr$^{-1}$, with minimum values of ~9 PgC yr$^{-1}$ and maximum of ~19 PgC yr$^{-1}$. Closer to the present, an average
of ~14 PgC yr$^{-1}$ is estimated, and extremes ranging from 9 PgC yr$^{-1}$ to 21 PgC yr$^{-1}$. However, the pattern of grouping of model results around an average value is much more noticeable, with the models INM-CM4-8, INM-CM5-0, MPI-ESM1-2-LR and UKESM1-0-LL reporting higher values than the others throughout the integration. Previously proposed values based on observations are 15.38 ± 3.47 PgC yr$^{-1}$ (years 2010-2019, Jones et al., 2023).

For NEP, there is also a temporal pattern of growth that intensifies after 1960 (Fig. 3c) In the initial years, the average values
are ~0.8 PgC yr$^{-1}$, with values between ~ -0.3 PgC yr$^{-1}$ and ~2.5 PgC yr$^{-1}$. Closer to the present, the average values are ~1 PgC yr$^{-1}$, with values between ~0 PgC yr$^{-1}$ and ~3.5 PgC yr$^{-1}$. It is noted that throughout the analyzed period the average values are always positive, and that it is also possible to notice a grouping of the results, with the EC-Earth3-Veg and GFDL-





ESM4 models reporting higher values than the others, throughout the integration. For reference, values based on observations in the period 2010-2019 are 1.51 ± 5.56 PgC yr$^{-1}$ (Jones et al., 2023).

Finally, NBP ranges around neutral balance (+/- ~1.5 PgC yr$^{-1}$) throughout most of the last century, with slight increase to positive values after 1960 (Fig. 3d). From the 1960s onwards, the mean values changed to ~0.25 PgC yr$^{-1}$, with extremes ranging from -1.5 to 2.5 PgC yr$^{-1}$. In the case of this variable, no clustering of the results was observed. For reference, values between the years 2010-2019 based on the CARDAMOM tool are 1.64 PgC yr$^{-1}$ (Bloom et al., 2016), and 0.7 ± 3.4 PgC yr$^{-1}$ and 0.083 ± 0.55 PgC yr$^{-1}$ based on top-down and bottom-up estimates, respectively (Jones et al., 2023).


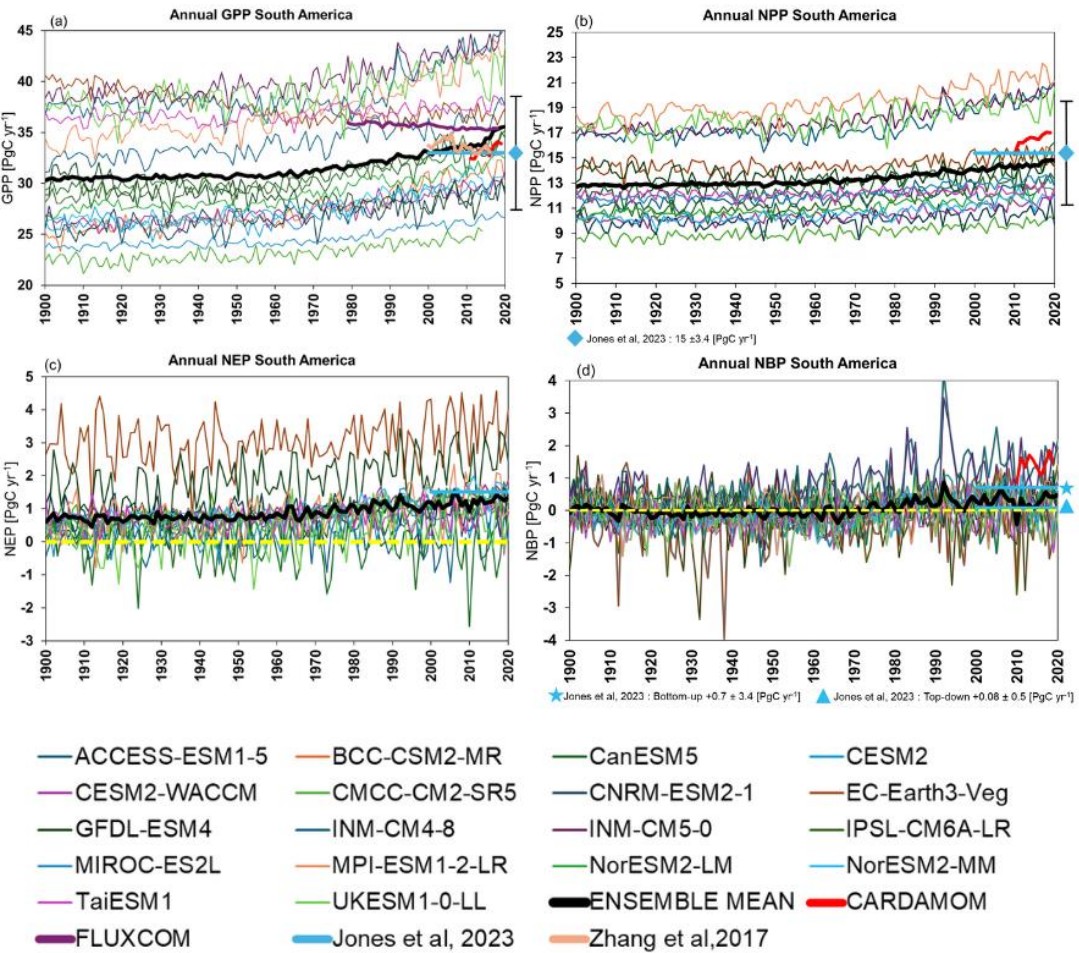

**Figure 3.** Total annual productivity fluxes computed over the entire South American continent, for each model (thin colored lines) and ensemble of model estimates (thick black line), for (a) Gross Primary Productivity of Carbon; (b) Net Primary Productivity of Carbon; (c) Net Ecosystem Productivity; and (d) Net Biome Productivity. Some previously reported values are also shown: GPP and NBP estimates by CARDAMOM (thick red lines); GPP by Zhang et al. 2017 (thick orange line); and the estimates by Jones et al. 2023 (blue symbols at the end of the periods).





Figure 4 shows cumulative annual NBP for the continent, as simulated from the models from 1900 to 2020. A decreasing trend is observed in the ensemble of the models (thick black line) until the mid-1970s, reaching a minimum of -4.7 PgC yr$^{-1}$. After this period, this trend reverses, and in the mid-1990s it becomes positive, reaching 7.4 PgC yr$^{-1}$ at the end of the

studied period (Fig. 4). Throughout the studied period, different temporal patterns can be observed among the results. A subset of models (ACCESS-ESM1-5, CNRM-ESM2-1, CMCC-CM2-SR5, INM-CM4-8, INM-CM5-0) presents values that are always positive and with an increasing trend, where the last two reach a accumulated maximum of 90 PgC in 2020. Another subset (CanESM5, GFDL-ESM4, TaiESM1, EC-Earth3-Veg) presents values that are always negative and decreasing, where the last model estimates a minimum of -30 PgC. The MPI-ESM1-2-LR and UKESM1-0-LL models

present values close to neutrality throughout the period. The IPSL-CM6A-LR and MIROC-ES2L models present negative values until the mid-1990s, and after these years they reach positive values with a final 8.5 PgC accumulated in 2020. A last group (CESM2, CESM2-WACCM, NorESM2-LM, NorESM2-MM) also estimate always negative values, but with a minimum in the mid-1970s and reversal after these years, ending with values between -11 and -4 PgC.

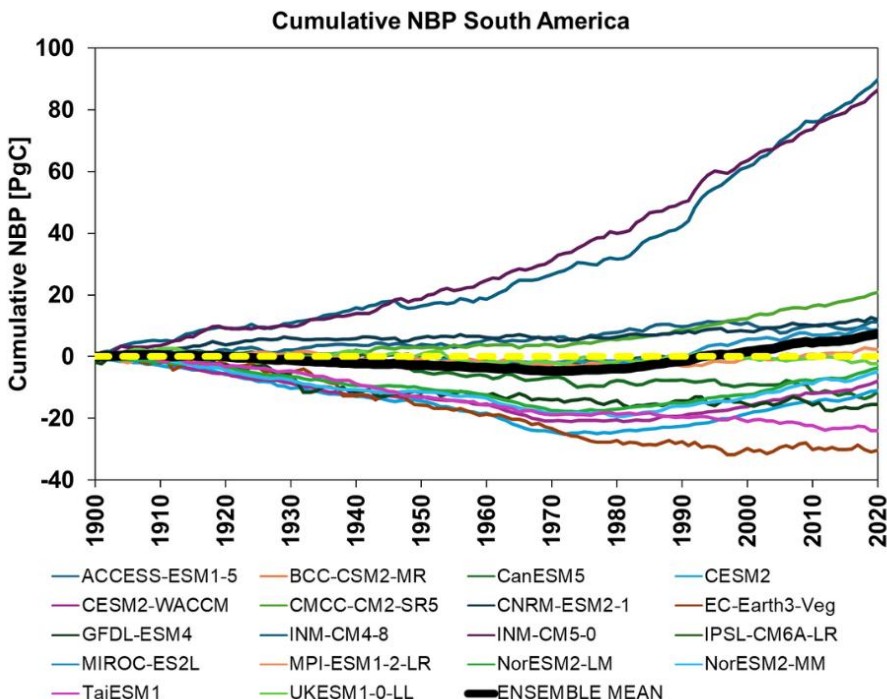

**Figure 4.** Cumulative annual NBP for the South American continent, as simulated from the models for the period 1900 to 2020.


Figure 5 shows the estimates of heterotrophic (Rh) and ecosystem (Reco) respiration terms, computed yearly over South America. Similar to the productivity terms, the respiration terms also present a temporal pattern with an increasing trend after 1960. Initially, Rh average values are ~11 PgC yr$^{-1}$, with minimums of ~8 PgC yr$^{-1}$ and maximums of ~17 PgC yr$^{-1}$





(Fig. 5a). Closer to the present, an average of ~12.5 PgC yr$^{-1}$ is observed, and extremes ranging from ~9 PgC yr$^{-1}$ to 20 PgC
yr$^{-1}$. As with NPP, Rh presents a noticeable pattern of clustering of models around the ensemble value, with the CanESM5
and UKESM1-0-LL models reporting higher values than the others throughout the integration. For reference, values between
the years 2010-2019 based on the CARDAMOM tool vary around 16.6 PgC yr$^{-1}$ (Bloom et al., 2016).

Reco averages ~29 PgC yr$^{-1}$, with minimum values of 22 PgC yr$^{-1}$ and maximum values of 38 PgC yr$^{-1}$ (Fig. 5b). Closer to
the present, an average of ~33 PgC yr$^{-1}$ is observed, and ranges from 26 PgC yr$^{-1}$ to 43 PgC yr$^{-1}$. It is also possible to note
that there are two groups of results, one with values always above the average, and another with values close to or below the
average of the models. For reference, previous literature estimates are 27 PgC yr$^{-1}$ for 1979-2018 (FLUXCOM, Jung et al.,
2020), and 36 PgC yr$^{-1}$ for the years 2010-2019, estimated by the CARDAMOM tool (Bloom et al., 2016).

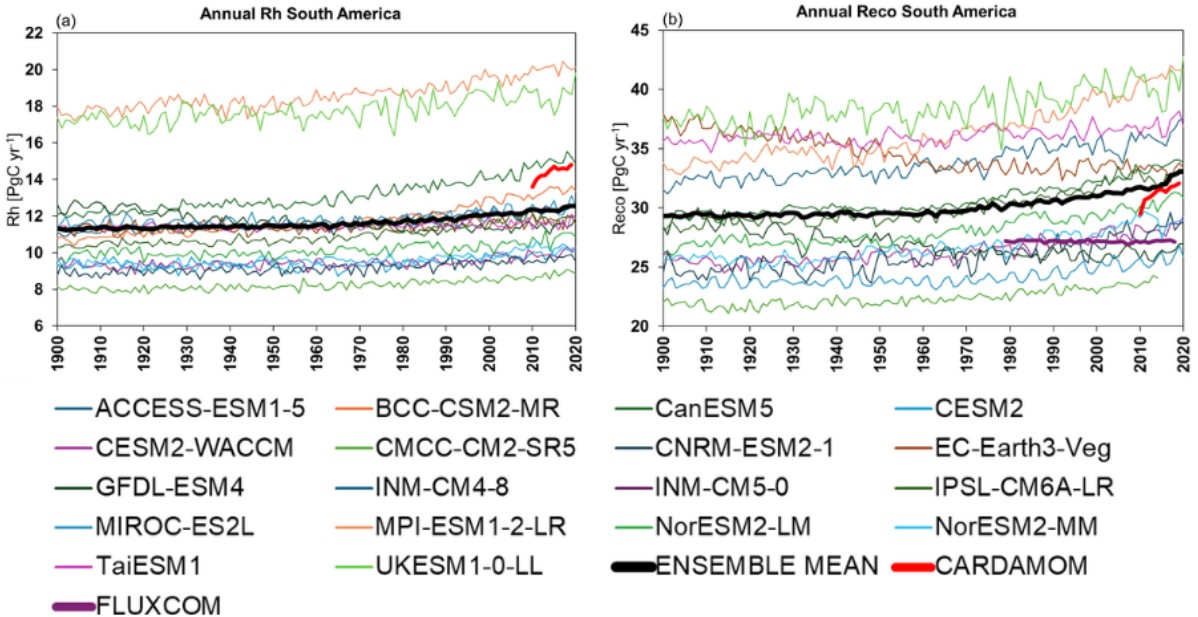

**Figure 5.** Total annual respiratory fluxes computed over the entire South American continent, for each model (thin colored lines) and
ensemble of model estimates (thick black line), for (a) Heterotrophic (Rh) and (b) Ecosystem (Reco) respiration terms. Estimates by
CARDAMOM (thick red line) and FLUXCOM (think purple line) are also shown.

The carbon stocks in vegetation (Cveg, Fig. 6a) and Total Carbon (the sum of carbon stock in vegetation and soil, Fig. 6b)
estimated for the entire South American continent show a similar behavior in their average. In the 1960s, the average went
from decreasing values to a gradual increase in the decades that followed until the recent period. The range of Cveg and
Total Carbon starts at ~71 PgC and ~200 PgC and ends at ~100 PgC and ~530 PgC, respectively. For reference of CVeg, the
values already proposed based on observations are 94.1 +/- 37.3 PgC yr$^{-1}$ for the years 2000-2019 (Jones et al., 2023) and
120 PgC for 1995-2005 (Gloor et al., 2012). For Total Carbon, 314.7 +/- 76 PgC yr$^{-1}$ was found for the years 2000-2019
(Jones et al, 2023).



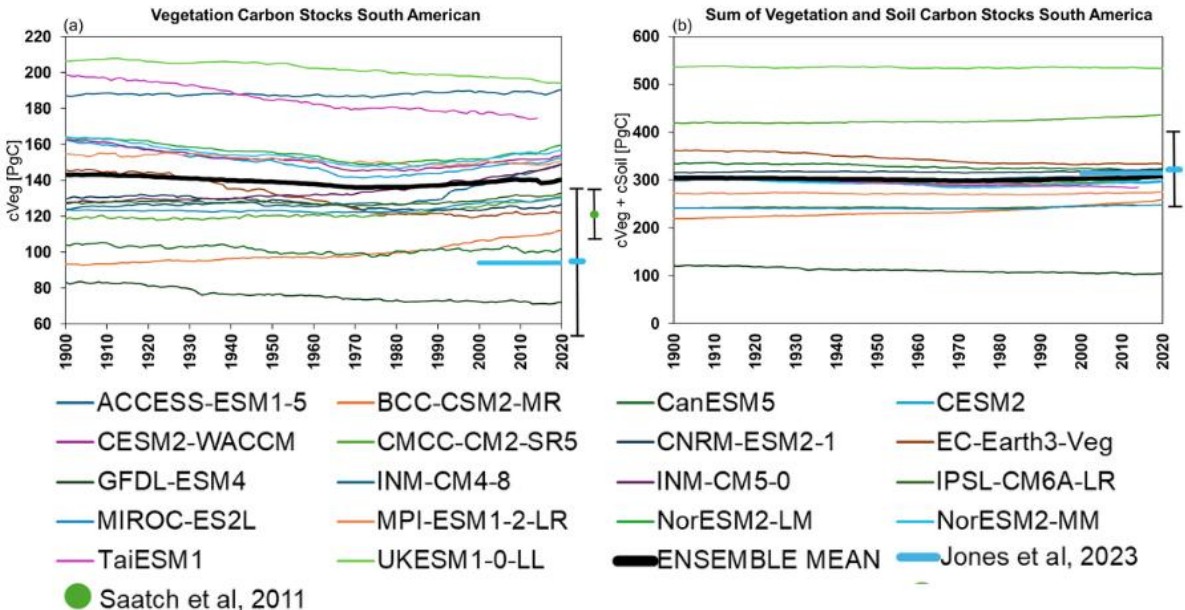

**Figure 6.** Carbon stock computed annually over the entire South American continent, for each model (thin colored lines) and ensemble of model estimates (thick black line), for (a) biomass vegetation only and (b) sum of vegetation and soil carbon stocks.

## 3.2 Amazonian and Savanna biomes

For a more detailed look at biomes levels, Figure 7 shows cumulative annual NBP calculated over the Amazonian and Savannas domains. In Amazonia, the cumulative NBP shows a positive signal trend over the studied period in 17 of the 18 models studied. CMCC-CM2-SR5 and INM-CM (versions 4-8 and 5-0) stand out, with the highest productivity estimates. At the end of the period, the ensemble of the models estimates an accumulated value of 14.8 PgC, with models varying between -4 PgC and ~52 PgC.

Computed over the savanas domain, accumulated NBP presents a range at the end of the studied period between -15 and 14 PgC among the models. There is a trend in the ensemble curve and in 14 of the 18 models, from neutrality to a negative signal throughout the studied period. Three different temporal patterns are observed. First, a subset formed by the models INM-CM4-8, INM-CM5-0 stands out with positive signals and a positive growth trend. A second subset presents values that are always negative and with a negative growth trend (CESM2, CESM2-WACCM, EC-Earth3-Veg, MPI-ESM1-2-LR,

NorESM2-LM, NorESM2-MM, TaiESM1). The third subset of models are around neutral balance, close to the ensemble line, without a clear trend of which is the direction of growth.





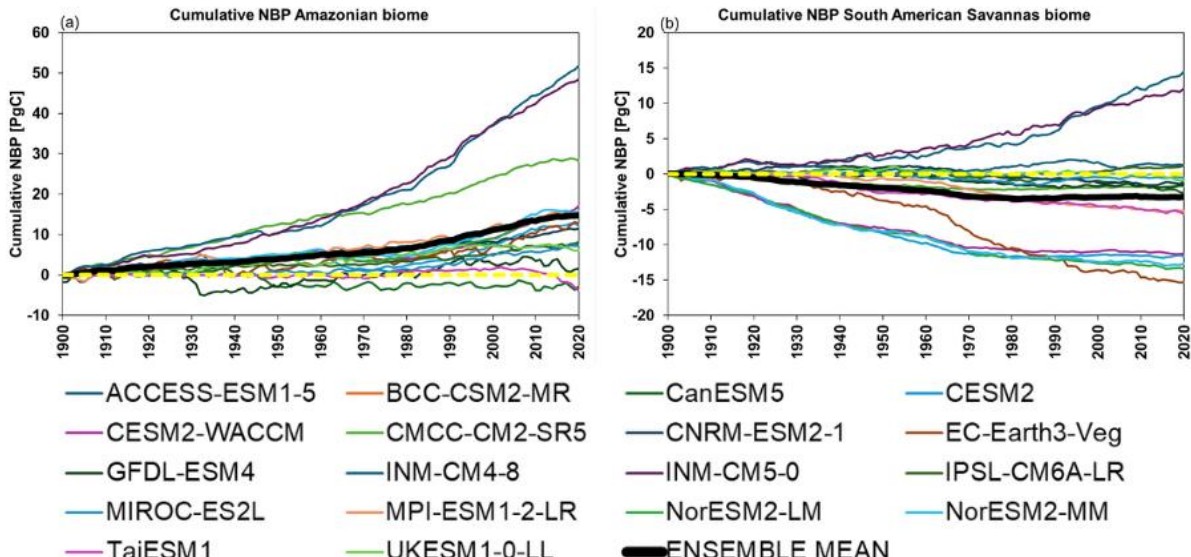

**Figure 7.** Cumulative annual NBP estimated by each model (thin colored lines) and ensemble for all models (thick black line) computed in the period from 1900 to 2020, computed over (a) Amazonia biome and (b) South American savannas domain.


Figure 8 shows boxplots of the carbon balance fluxes in the Amazon region as simulated by each model, for years classified as either widespread below-average or above-average precipitation (referred to as 'generally dry' or 'generally wet', respectively). Considering all models, NBP values ranged from ~-3 to 1.5 PgC yr$^{-1}$ (Fig. 8a). In years where most of the territory had below-average precipitation values, the models generally estimated net fluxes predominantly negative or close

to zero (<0.25 PgC yr$^{-1}$). In years marked by above-average precipitation, these values were always positive, and in some cases close to zero (>0.01 PgC yr$^{-1}$). Individually, each model always estimated NBP values lower (higher) for dry (wet) years. For the first quartile (Q1) in wet years, the models estimated positive values up to 0.75 PgC yr$^{-1}$, with the exception of the MIROC-ES2L model (-0.07 PgC yr$^{-1}$). In dry years, Q1 was generally >-0.25 PgC yr$^{-1}$, with the CanESM5, EC-Earth3-Veg and GFDL-ESM4 models estimating values between -1 and -1.5 PgC yr$^{-1}$, and INM-CM models close to zero. The third

quartile (Q3) in wet years is always >0.25 PgC yr$^{-1}$, with EC-Earth3-Veg, INM-CM4-8 and INM-CM5-0 models estimating values close to 1 PgC yr$^{-1}$. Even in dry years, Q3 were positive between 0.1 and 0.4 PgC yr$^{-1}$ in 9 models. The remaining models estimated Q3 values for dry years between -0.2 and -0.5 PgC yr$^{-1}$. For reference, the values estimated in previous studies in Amazonia are ~0.4 PgC yr$^{-1}$ for 1980-1994 (Malhi et al., 2021), between 0.11 and 0.21 PgC yr$^{-1}$ for the period 2000-2010, between -0.31 and 0.01 PgC yr$^{-1}$ for 2010 (drought year) (Aragão et al., 2014), and between 0.88 and 1.16 PgC

yr$^{-1}$ for the years 2010-2019 estimated by the CARDAMOM tool (Bloom et al., 2016).

For GPP in the Amazon region, most models reported values between 11 and 20 PgC yr$^{-1}$ (Fig. 8b). Three models (INM-CM4-8, INM-CM5-0, TaiESM1) estimated higher values between 20 and 25 PgC yr$^{-1}$. Individually, almost all models reported lower GPP for dry years than in wet years. However, in a small number of cases, they did not follow this pattern and reported values that were higher in dry years. Specifically, four models (BCC-CSM2-MR, CESM2-WACCM, MIROC-



ES2L, NorESM2-LM) estimated Q3 values in dry years equal to or higher than those estimated in wet years. For reference, the values reported in the literature are 21 PgC yr$^{-1}$ (Malhi et al., 1999), 19 PgC yr$^{-1}$ for 1979-2018 (FLUXCOM, Jung et al., 2020), and 18 PgC yr$^{-1}$ for the years 2010-2019, estimated by the CARDAMOM tool (Bloom et al., 2016).

In general, for Rh, 16 models remained between the extremes of 3.7 to 6.7 PgC yr$^{-1}$ and two models (MPI-ESM1-2-LR, UKESM1-0-LL) settled above 7 PgC yr$^{-1}$, reaching up to 9 PgC yr$^{-1}$ (Fig. 8c). Individually, 13 models estimated the upper

quartile (Q3) for dry years, equal to or lower than Q3 for wet years. As a reference, the Rh value estimated by the CARDAMOM tool (Bloom et al., 2016) is ~7 PgC yr$^{-1}$ (2010-2019) and 6.5 PgC yr$^{-1}$ by Malhi et al., (1999).

Finally, 15 models estimate between 0 and 0.5 PgC yr$^{-1}$ of emissions due to disturbances in the Amazon (Fig. 8d), and 3 models (EC-Earth3-Veg, GFDL-ESM4, TaiESM1) estimate between 0.5 and 2.5 PgC yr$^{-1}$. Models BCC-CSM2-MR, CESM2, INM-CM4-8, INM-CM5-0 did not present results. In predominantly dry years, most models estimate the median

equal to or greater than the results in wet years, only the models IPSL-CM6A-LR and MPI-ESM1-2-LR did not follow this behavior. As a reference, the value of emissions due to disturbances estimated by the CARDAMOM tool (Bloom et al., 2016) is ~0.4 PgC yr$^{-1}$.

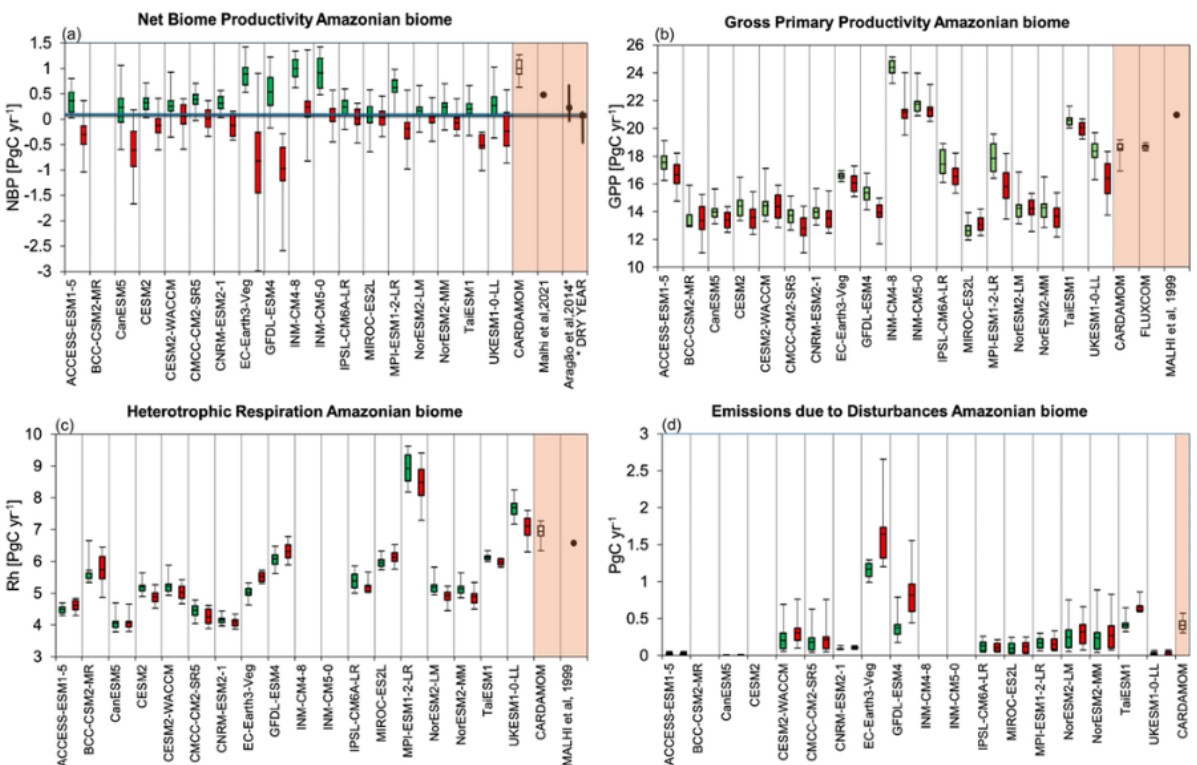

**Figure 8.** Boxplots of the carbon balance fluxes ( a) Net Biome Productivity, b) Gross Primary Productivity, c) Heterotrophic respiration, d) emissions due to disturbances) in the Amazon region as simulated by each model, for years classified as either widespread below-average ("dry years", red boxes) or above-average precipitation ("wet" years, green boxes). Shaded areas of the figures show estimates found in the literature.



Figure 9 shows boxplots similar to Fig. 8, but for the fluxes in the South American Savannas domain. In these biomes, NBP estimates range from ~-1 to 0.7 PgC yr$^{-1}$ (Fig. 9a), with medians predominantly negative or close to zero, in years where at least 2/3 of the biome had below-average precipitation. In years marked by above-average precipitation, these values were near zero or small positive, reaching up to 0.4 PgC yr$^{-1}$ for the INM-CM4-8 model. Individually, the median found for each model always had higher NBP values for wet years than for dry years. Q1 in dry years ranged from -0.1 to -0.6 PgC yr$^{-1}$,

while it was between -0.1 and 0.4 PgC yr$^{-1}$ for wet years. Q3 in wet years was always greater than 0 PgC yr$^{-1}$, with the INM-CM4-8 and INM-CM5-0 models estimating values close to 0.5 PgC yr$^{-1}$. In dry years, Q3 was close to 0.1 PgC yr$^{-1}$ by INM-CM4-8, reaching -0.2 PgC yr$^{-1}$ in the EC-Earth3-Veg model. For reference, previous estimates are between 0 and 0.37 PgC yr$^{-1}$ for the years 2010-2019, by the CARDAMOM tool (Bloom et al., 2016).

For GPP in the Savannas, 7 models reported values between 2 and 5 PgC yr$^{-1}$ and 11 models reported values between 5 and

8.5 PgC yr$^{-1}$ (Fig. 9b). Individually, 14 models reported median GPP equal to or lower for dry years than in wet years. However, in a small number of cases, they did not follow this pattern and reported values that were higher in dry years. Specifically, 3 models (CanESM5, CNRM-ESM2-1, MIROC-ES2L) estimated Q3 values in dry years equal to or higher than in wet years. For reference, values previously reported are 5 PgC yr$^{-1}$ for 1979-2018 (FLUXCOM, Jung et al. 2020), and 4 PgC yr$^{-1}$ for 2010-2019, estimated by the CARDAMOM tool (Bloom et al., 2016).

For Rh (Fig. 9c) , 13 models remained between 1 and 2.5 PgC yr$^{-1}$ and 3 models (CanESM5, MPI-ESM1-2-LR, UKESM1-0-LL) settled above 3 PgC yr$^{-1}$ to 4 PgC yr$^{-1}$. Individually, 9 models estimated the upper quartile (Q3) for dry years, equal to or lower than Q3 for wet years. As a reference, Rh estimated by the CARDAMOM tool (Bloom et al., 2016) is ~2.3 PgC yr$^{-1}$ (2010-2019).

The emissions due to disturbances were estimated between 0 and 0.4 PgC yr$^{-1}$ (Figu. 9d) for 15 models and between 0.5 and

1 PgC yr$^{-1}$ for 3 models. Models BCC-CSM2-MR, CanESM5, CESM2, INM-CM4-8, INM-CM5-0 did not present results. In predominantly dry years, 9 models estimate the median equal to or greater than the results in wet years, and only MPI-ESM1-2-LR and NorESM2-LM did not follow this behavior. As a reference, emissions due to disturbances estimated by the CARDAMOM tool (Bloom et al., 2016) is ~0.1 PgC yr$^{-1}$.



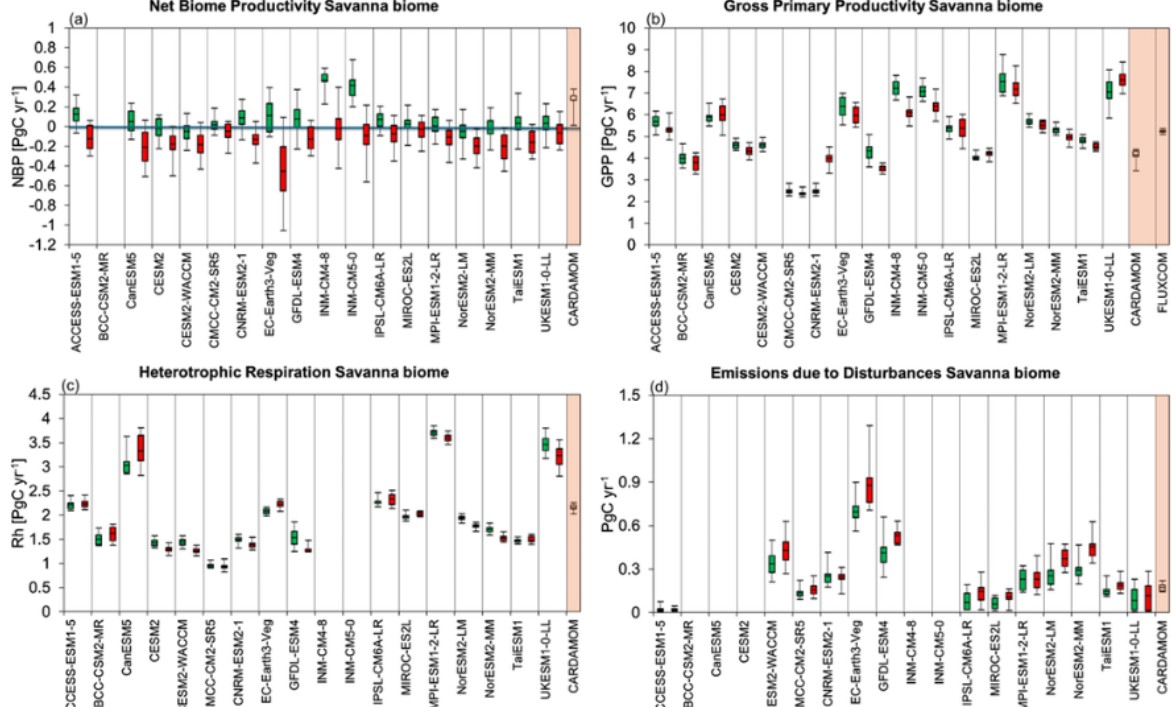

**Figure 9.** Same as figure 8, but for the fluxes computed in the domains of South American Savannas.


Summarising results in Fig. 10, we show the boxplots of annual values computed for the ensemble of the models, for wet (green boxes) and dry (red boxes) years. NBP (Fig. 10a) were established between -1 and 1 PgC yr$^{-1}$ for Amazonia and between -0.5 and 0.4 PgC yr$^{-1}$ for Savannas. In years where precipitation values were above average, NBP is generally positive for both biomes. Conversely, in years marked by below-average precipitation, NBP is generally negative. The values observed in the literature are close to those calculated in this work.

In the Amazon region, GPP ranges between 15 and 17 PgC yr$^{-1}$ (Fig. 10b), while for the Savannas the values were between 4.8 and 5.5 PgC yr$^{-1}$. In both biomes, the average GPP is lower for dry years than in wet years.

Rh is also higher for the Amazon biome, between 5.3 and 5.7 PgC yr$^{-1}$, than for the Savannas, between 1.9 and 2.1 PgC yr$^{-1}$ (Fig. 10c). And, as for GPP, both biomes reported higher values for wet years than for dry years, but with smaller differences.

For emissions from disturbances (Figu. 10d), the ensemble estimates were similar for both biomes, between 0.2 and 0.4 PgC yr$^{-1}$. In predominantly dry years, the values were higher than in wet years for both biomes. As a reference, the value of emissions from disturbances estimated by the CARDAMOM tool (Bloom et al., 2016) is ~0.4 PgC yr$^{-1}$ for the Amazon Biome, a value higher than the ensemble of the models, and ~0.1 PgC yr$^{-1}$, lower than the average for the Savannas biomes.






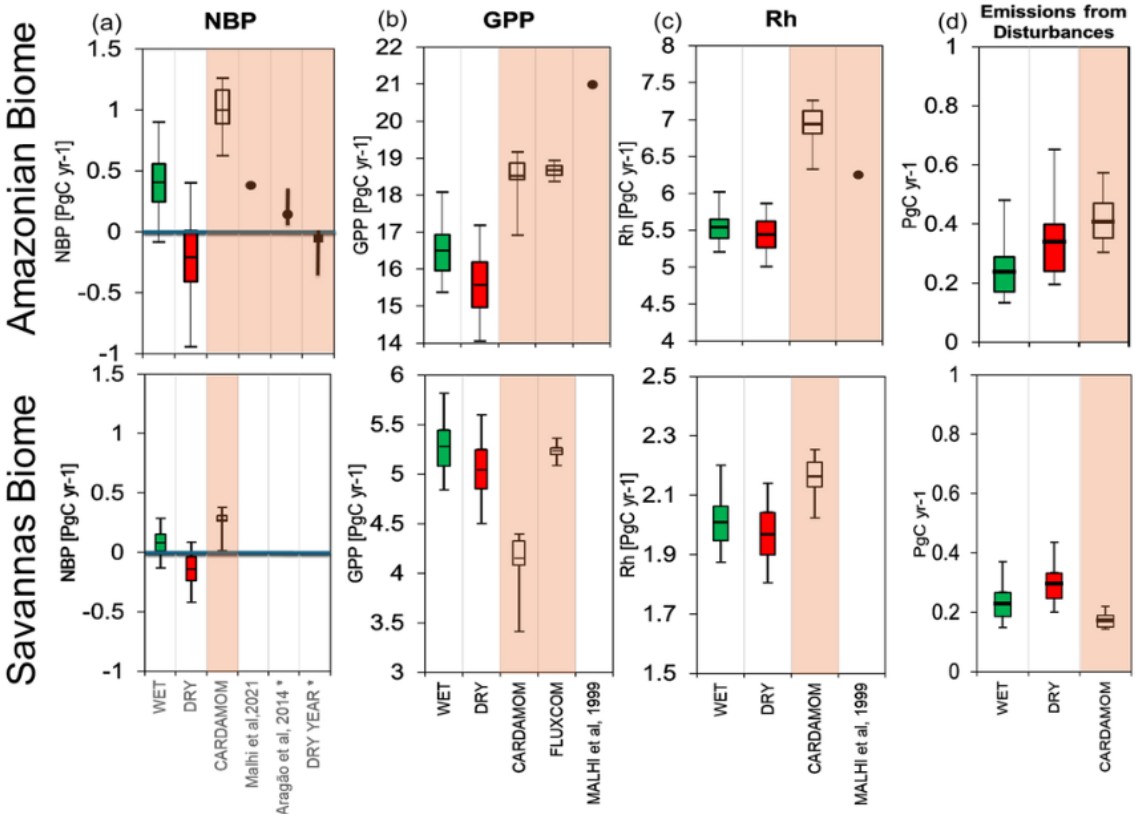

**Figure 10.** Boxplots of annual values computed for the ensemble of the models, for wet (green boxes) and dry (red boxes) years.

## 4. Discussions

### 4.1 Role of South American biomes in the global carbon cycle

South America has substantial relevance within the contemporary global carbon cycle (Table 2). Taking into account the

analyses presented here and data from other studies, it is possible to assess the role that the main carbon fluxes and stocks on

the continent have in relation to the global totals. For GPP, between 25% and 30% of the global total of this flux is located

on the continent. In the case of NPP, the values are between 21% and 28% of the global total. The NEP and NBP on the

continent are also significant, and amount to between 17% and 50%, and between 15% and 30%, respectively, of the total

verified globally. In addition, Ra and Rh contribute with ~44% and between 16% and 31%, respectively, of their global

totals. Carbon stocks in vegetation and soil are also significant, with 36% of all cVeg and 11% of all cSoil located within

South America. The amounts presented above represent the ratio between the annual ensembles across all models for the

South American continent between the years 2000-2020 (Table 2, Estimates for South America, CMIP models) and the

global values reported by other studies (Table 2, Global Estimates). It is also interesting to note that there is no relevant



discrepancy between the values calculated in this study, based on the CMIP6 models, and those previously reported by other
authors, obtained by various methods, including computational models, field experiments, and remote sensing.

The temporal evolution of continental NBP shown in Fig. 4 shows that models do not always agree in their simulations and, cumulatively, results in a large spread after a few years of simulation. Most models initially present values close to neutrality, and at the end of the period, after 1990, positive values. In the intermediate years, between 1930-1990, the ensemble of the models presented negative values. In general, it can be noted that the temporal evolution of these NBP
values indicate a combination of the values estimated for the Amazon and the Savannas. A group of models (CESM2, CESM2-WACCM, NorESM2-LM, NorESM2-MM, MIROC-ES2L) shows values that follow the ensemble pattern, and it is interesting that almost all of these models have the CLM5 as their land surface component. Lawrence et al., (2019) highlights that this update of the CLM has better performance than previous versions. Hajima et al., (2020) presented the improvements in the MIROC-ES2L ESM, including the explicit incorporation of carbon-nitrogen interactions in the
terrestrial component. Another group of models (CanESM5, EC-Earth3-Veg, TaiESM1, GFDL-ESM4) presents different and decreasing values from the ensemble curve, which can potentially be associated with a small influence of the $CO_2$ fertilization effect in these models (Table 1) (Lee et al., 2020 e Döscher et al., 2022).

Another interesting feature in the temporal evolution of cumulative NBP for South America are the values reported by the INM-CM4-8 and INM-CM5 models, which are substantially higher than the others throughout the studied period, linked to
the fact that these models also estimate high GPP values (Fig. 3a), which was also noted by Gier et al., (2024), and the fact that these models do not consider the dynamics of disturbances due to fire and land use change (Table 1, Figs. 8d and 9d). Despite the patterns identified here, it is important to emphasize that NBP represents the net budget, the difference between the components of C uptake by photosynthesis and emissions by respiration and disturbances. These components may present similar magnitudes and small numerical differences; thus, relatively small errors in any component can be
propagated into large errors in the NBP values (Table 2, Jones et al., 2023).







**Table 2.** Global and South American Estimates of Carbon Cycle Fluxes and Stocks.

| VARIABLE | GLOBAL ESTIMATES | | | SOUTH AMERICA ESTIMATES | | |
|---|---|---|---|---|---|---|
| | REFERENCE | PERIOD | VALUE [PgC yr⁻¹] | REFERENCE | PERIOD | VALUE [PgC yr⁻¹] |
| GPP | JONES et al, 2023<br>CHEN et al, 2017<br>ZHANG et al, 2017 | 2010-2019<br>1971–2010<br>2000–2016 | GPP = 131<br>GPP = 112<br>GPP = 127 | JONES et al, 2023<br>CMIP6 models (Table 1) | 2010-2019<br>2000-2020 | GPP = 33 ± 5.8<br>GPP = 33 ± 0.8 |
| NPP=GPP-Ra | CIAIS et al, 2021<br>JONES et al, 2023 | 2000-2010<br>2010-2019 | NPP = 50 (57,44)<br>NPP = 65 | JONES et al, 2023<br>BLOOM et al, 2016<br>ZHAO et al, 2006<br>CMIP6 models (Table 1) | 2010-2019<br>2000-2010<br>2000-2010<br>2000-2020 | NPP = 14 ± 3.5<br>NPP = 13 ± 4.4<br>NPP = 13 ± 2.7<br>NPP = 14 ± 0.3 |
| NEP=-NEE=NPP-Rh | CIAIS et al, 2021<br>JONES et al, 2023 | 2000-2010<br>2010-2019 | NEP = 2.4 ± 6.2<br>NEP = 7.8 * | JONES et al, 2023<br>CIAIS et al, 2021<br>CMIP6 models (Table 1) | 2010-2019<br>2000-2010<br>2000-2020 | NEP = 1.2 ± 4.8<br>NEP = 0.7 ± 2.9<br>NEP = 1.2 ± 0.1 |
| NBP=NEP-disturbances | JONES et al, 2023<br>CHANG et al, 2017 | 2010-2019<br>1971–2010 | NBP = 2<br>NBP = 1 ± 0.4 | CMIP6 models (Table 1) | 2000-2020 | NBP = 0.3 ± 0.2 |
| Ra | TANG et al, 2019<br>SCHLESINGER et al, 2000 | 1980-2012<br>anos 1980 | Ra = 43.8 ± 0.4<br>Ra = 45 | CMIP6 models (Table 1) | 2000-2020 | Ra = 19.3 ± 0.6 |
| Rh | CIAIS et al, 2021<br>JIAN et al, 2022<br>SCHLESINGER et al, 2000 | 2000-2010<br>anos 1980 | Rh = 39 (32,45)<br>Rh = 68<br>Rh = 75 | CIAIS et al, 2021<br>CMIP6 models (Table 1) | 2000-2010<br>2000-2020 | Rh = 12,3 (10.1,14.4)<br>Rh = 12.3 ± 0.1 |
| Reco=Ra+Rh | JONES et al, 2023 | 2010-2019 | Re = 123,2 | CMIP6 models (Table 1) | 2000-2020 | Re = 31.8 ± 0.6 |
| VARIABLE | REFERENCE | PERIOD | VALUE [PgC] | REFERENCE | PERIOD | VALUE [PgC] |
| cVeg | JONES et al, 2023 | 2010-2019 | cVeg = 384 ** | GLOOR et al, 2012<br>JONES et al, 2023<br>CMIP6 models (Table 1) | 1995-2000<br>2010-2019<br>2000-2020 | cVeg = 120<br>cVeg = 140 ± 29.4<br>cVeg = 139 ± 0.7 |
| cSoil | JONES et al, 2023<br>SCHLESINGER et al, 2000 | 2010-2019<br>anos 1980 | cSoil = 1400<br>cSoil = 1500 ** | CMIP6 models (Table 1) | 2000-2020 | cSoil = 165 ± 1.8 |

\* NO INFORMATION FROM EUROPE AND AFRICA          \*\* NO INFORMATION FROM EUROPE


### 4.2 Comparative responses of Amazonia and Savannas

Apart from the magnitude of the fluxes, we see clear similarities of Amazonia and Savannas ecosystems responses when widespread dry years occur (Fig. 8 to 10), and these responses drive most of the interannual variability of the whole continent. However, it is worth analysing the slight differences between these two ecosystem types, as they represent the

most comprehensive of the continent.

The temporal evolution of the cumulative NBP for the Amazon (Fig. 7) presents positive and increasing values throughout the contemporary period for the vast majority of models; the ensemble curve follows a similar characteristic, highlighting a greater slope from the 1970s onwards. One model (CanESM5) differs from this trend, with values always negative, and the set of models GFDL-ESM4, TaiESM1, UKESM1-0-LL, present a decreasing trend from the 1990s onwards. As for South

America, the INM-CM family stands out positively in its NBP results; this characteristic is also noted even in dry years, in which the lower quartile of NBP of these models is positive (Fig. 8a) and they are also the ones that present the highest GPP values (Fig. 8b).

This positive signal from most models may be associated with a response to the increase in atmospheric $CO_2$ concentration, which enhances the fertilization effect on the photosynthesis process. This influence on the Amazon has been evaluated in

several modelling studies such as Lapola et al., (2009), Huntingford et al., (2013), and Sampaio et al., (2021), showing that



fertilization acts on both the spatial distribution and the resilience of forests. In fact, the global atmospheric concentration of this gas has increased significantly, notably after the 1960s (Quirk et al., 2021); data from the CDIAC (Carbon Dioxide Information and Analysis Center) show this signal in more than 65% of the 4,018 stations spread across the globe (Gilfillan et al., 2021 and Friedlingstein et al., 2023).

As for the savannas, where the Brazilian Cerrado represents most of its territorial extension, there is a predominance of negative values in cumulative NBP (Fig. 7b), except in the INM-CM4-8 and INM-CM5-0 models, which, as in the other regions, persist with strong positive values throughout the studied period. On the negative side of NBP, the models CESM2, CESM2-WACCM, EC-Earth3-Veg, NorESM2-LM, NorESM2-MM stand out, whose coupled surface models are CLM in its version 5. Lawrence et al., (2019) highlighted that this version of the CLM model brought improvements in the carbon and

nitrogen cycle, an improved treatment in $CO_2$ fertilization, and an update of the stomatal physiology.

Another factor contributing to the negative NBP in Savannas (Fig. 7b), especially in Brazil, is the history related to land use change. Since the 1960s, several policies and initiatives have been created by the Brazilian State with the aim of encouraging the expansion of agriculture in the region. Programs such as POLOCENTRO (Cerrados Development Program), created to stimulate occupation through development and modernization, and PRODECER (Japanese-Brazilian Program for

Agricultural Development of the Cerrados Region), a cooperation agreement with Japan, played an important role in adapting cultivars and management techniques for the region (Marciel et al., 2017). Among the consequences portrayed is the transformation of the landscape, exchanging native vegetation for grain monocultures in large areas, in addition to the loss of biodiversity, contamination of water resources by pesticides, and soil degradation (Silva et al., 2023). This anthropogenic process coincides with large-scale changes in atmospheric circulation. As described in Hofmann et al. (2023),

there was an increase in subsidence over the Cerrado, causing a decrease in rainfall. A possible explanation was proposed by Jones and Carvalho (2018) and Reboita et al, (2019), where an expansion of the Hadley cell to the south was noted, which cooperated with an intensification in the South Atlantic Subtropical Anticyclone (ASAS), reducing the transport of moisture from the Atlantic Ocean to the continent. As a consequence, from the 1960s onwards, an increase in near-surface temperature and a reduction in the frequency of rainy days during the dry season and the beginning of the rainy season were

observed in the central part of the South American continent (Hofmann et al., 2021).

Although the total emissions due to disturbances estimated in both biomes are similar (Fig. 10d), these values reflect a larger extent of the Amazonian domain compared to the Savannas. When looking at the average emissions per hectare, the 18 ESMs present values for wet years around 0.37±0.1 MgC ha-1 yr-1 in the Amazon, and 0.72±0.12 MgC ha-1 yr-1 in the Savannas, while for dry years 0.52±0.1 MgC ha-1 yr-1 in the Amazon and 0.93±0.12 MgC ha-1 yr-1 in the Savanna areas.




### 4.3 Overall agreement between models and reference data

An interesting feature in our analyses is an overall agreement between model results and reference data in several sets of the results. For example, looking at the annual values of GPP, NPP, NEP, NBP (in Fig. 3a-d), and respiration terms Rh and Reco (Fig. 5a,b) for the whole of South America, we note that the average values of these fluxes, considering all model estimates, compare relatively well with the reference data in most cases. That is also the case when observing the estimates from the specific models and subregions in separated, as shown in Fig. 8 and 9, also including information for disturbances, when

computed. Relative values between wet and dry years also seem to make sense in these results, with higher NBP and GPP in wet years (Fig. 8a-b,9a-b), and higher Rh and disturbances in dry years (Fig. 8c-d,9c-d), which is easier to see in Fig. 10 for both Amazonia and Savannas. That is important to consider when discussing results from CMIP6 models, to a great extent, indicating accuracy and supporting most model results analysed here.

### 4.4 Future

Our analyses can help us to quantify the relative impacts of dry and wet years on important elements of the C cycle. Together with results from other studies that estimate future climate in South America, we can infer some of the changes that may occur in the ecosystems of our study region. For example, we believe that the climate conditions projected for the 21st century may increase the vulnerability of the ecosystems in South America. In the Amazon biome, it is projected a reduction

in total annual rainfall, accompanied by more intense but less frequent rains, which will contribute to an increase of 5% to 50% in the frequency of severe droughts by 2030 (Marengo, 2008). In addition, the scenarios point to more prolonged and intense heat waves (Dos Santos et al., 2024). With the worsening of the drought and the increase in temperatures, tree mortality tends to grow, reducing GPP and intensifying the Reco. These processes may enhance emissions from disturbances and contribute to the reduction of the biome's net biome production (NBP).

In Savanna biomes, precipitation patterns may vary regionally, with an increase of the frequency of droughts and lengthening of heat waves, intensifying water deficit and climate risks (Almazroui et al., 2021). This scenario favors the increase in the frequency of fires and the degradation of vegetation, accelerating the emission of $CO_2$ and other greenhouse gases. Thus, a feedback loop may be established, where the greater release of carbon in the atmosphere raises the temperature, making extreme weather events even more frequent and severe. With these scenarios and in extreme situations,

parts of South America may change from carbon sink to source, compromising the global carbon balance (Ripple et al., 2023).





### 5. Conclusion

We evaluated the carbon cycle over South America, covering the Amazon biome and South American Savanna areas, using 18 Earth System Models (ESMs) participating in Phase 6 of the Coupled Model Intercomparison Project (CMIP6; Eyring et

al., 2016a), for the period from 1900 to 2020. We analysed the carbon fluxes and stocks that make up the carbon balance, assessed how well CMIP6 models represent these variables compared to previously published results, and proposed an update of carbon cycle estimates for the continent, considering the multi-model mean of all 18 models. Additionally, we examined the potential of the continent and its biomes to transition into carbon sources, given the intensification of climate change.

Several important aspects can be highlighted from our analyses. One is the importance of South America in the global contemporary carbon cycle, with substantial contributions to major fluxes and stocks (25-30% of the global GPP, 21-28% of total NPP, 17-50% of NEP, 15-30% of the NBP, ~44% of Ra, 16-31% of Rh, 36% of all cVeg, and 11% of all cSoil). The evolution of continental NBP reflects a combination of estimates for the Amazon and Savannas. Most models start with values close to neutrality, showing negative values between 1930 and 1990, and becoming positive after 1990. ESMs using

the latest version of CLM produced results consistent with the multi-model mean, while those with earlier CLM versions or without an interactive nitrogen cycle showed declining values, deviating from the ensemble curve. The INM-CM4-8 and INM-CM5 models report significantly higher NBP values compared to others throughout the studied period, likely due to their high GPP estimates (Fig. 3a) and the absence of fire and land use change dynamics (Table 1, Fig. 8d and 9d) (Gier et al., 2024). However, it is important to note that NBP represents the net balance between carbon uptake and emissions,

meaning that small errors in any component can lead to large discrepancies in NBP values (Table 2, Jones et al., 2023).

Another aspect is the overall pattern of results considering years that are predominantly dry or wet. For NBP, in dry years the models estimate values that are predominantly negative or close to zero (<0.25 PgC yr$^{-1}$ Amazonia, and <0.2 PgC yr$^{-1}$ Savannas), and in wet years these values are always positive, and in some cases also close to zero (>0.01 PgC yr$^{-1}$ Amazonia, 0-0.4 PgC yr$^{-1}$ Savannas). Almost all models reported lower GPP for dry years than in wet years in both biomes. However, in

a small number of cases in Amazonia (BCC-CSM2-MR, CESM2-WACCM, MIROC-ES2L, and NorESM2-LM) they did not follow this pattern and reported values that were higher in dry years. For Rh in both biomes, the estimated values are higher for wet years, and in general the emissions from disturbances were higher in dry years than in wet years in both biomes. Despite similar total emissions from disturbances in both biomes, per-hectare emissions are higher in the Savannas. In wet years, emissions average 0.37±0.1 MgC ha$^{-1}$ yr$^{-1}$ in the Amazon and 0.72±0.12 MgC ha$^{-1}$ yr$^{-1}$ in the Savannas, while

in dry years, emissions rise to 0.52±0.1 MgC ha$^{-1}$ yr$^{-1}$ in the Amazon and 0.93±0.12 MgC ha$^{-1}$ yr$^{-1}$ in the Savannas.

In regard to the temporal evolution of the cumulative NBP in the Savannas, particularly in Brazil, changes in land use and cover can be associated with the values estimated by the models, in particular by the ones that consider these processes (Fig. 7b). For example, deforestation and agricultural expansion since the 1960s (Marciel et al., 2017), which lead to important modifications of the land surface (Silva et al., 2023), also coincide with changes in atmospheric circulation, such as the



expansion of the Hadley cell and intensification of the South Atlantic Subtropical Anticyclone (ASAS), potentially caused reduced rainfall and increased temperatures (Hofmann et al., 2021) and fluctuations in the NBP values in the region.

Our analysis indicates an overall agreement between model results and reference data across multiple carbon fluxes. Annual estimates of GPP, NPP, NEP, NBP (Fig. 3a-d) and respiration terms (Rh and Reco) (Fig. 5a-b) for South America generally align well with reference values. This consistency is also observed when analysing specific models and subregions (Fig. 8-

9), including disturbance-related data when considered. Relative differences between wet and dry years follow expected patterns, with higher NBP and GPP in wet years (Fig. 8a-b, 9a-b) and higher Rh and disturbance levels in dry years (Fig. 8c-d, 9c-d), particularly evident in Fig. 10 for both Amazonia and Savannas. These findings highlight, to a great extent, the accuracy of CMIP6 model results and reinforce their reliability for analysing carbon cycle dynamics.

Our analysis quantifies the relative impacts of dry and wet years on key carbon cycle components. Based on projections for

the 21st century, South American ecosystems may become increasingly vulnerable to climate change. In the Amazon, decreasing annual rainfall, coupled with more intense but less frequent storms, is expected to increase severe drought frequency by 5% to 50% by 2030 (Marengo, 2008; Dos Santos et al., 2024). More prolonged and intense heat waves will exacerbate drought conditions, leading to higher tree mortality, reduced GPP, and intensified Reco, ultimately lowering the net biome production (NBP). In Savanna biomes, regional variations in precipitation and more frequent droughts and heat

waves will increase water deficits and climate risks (Almazroui et al., 2021). This will favor wildfires and vegetation degradation, accelerating $CO_2$ emissions and other greenhouse gases. A feedback loop may emerge, where higher carbon emissions raise temperatures, further intensifying extreme climate events. In extreme scenarios, parts of South America could shift from a carbon sink to a carbon source, compromising the global carbon balance (Ripple et al., 2023).

*Author contributions*. MBS led the writing and analysis of the paper. MFC assisted with conceiving and designing the study, implementing the analyses, and interpreting the results. CvR supervised the study. All authors contributed to the writing of the paper.

*Competing interests*.

The contact author has declared that none of the authors has any competing interests.

*Acknowledgements*. We acknowledge the World Climate Research Programme (WCRP), which, through its Working Group on Coupled Modeling, coordinated and promoted CMIP6. We thank the climate modeling groups (listed in Tab. 1) for producing and making available their model output, the Earth System Grid Federation (ESGF) for archiving the data and

providing access, and the multiple funding agencies that support CMIP and ESGF. The computational resources of the National Institute for Space Research (INPE, Brazil) were used to compute these results and are gratefully acknowledged. We thank Luke Smallman for his support in producing the CARDAMOM analyses.



*Financial support*. CDJ was supported by the Joint UK BEIS/Defra Met Office Hadley Centre Climate Programme
(GA01101) and the European Union's Horizon 2020 research and innovation programme under Grant Agreement No
101003536 (ESM2025 - Earth System Models for the Future). MBS was supported by the PROEX (Academic Excellence
Program, Brazil), process number 88887.6140952021-00. This study was financed in part by the *Coordenação de
Aperfeiçoamento de Pessoal de Nível Superior - Brasil (CAPES)* - Finance Code 001 and had institutional support from the
Graduate Program in Earth System Science (PGCST) at INPE, Brazil. MFC acknowledges support from the São Paulo
Research Foundation (FAPESP, Brazil, Process 2015/50122-0).

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
