# Peer review of "Assessing terrestrial carbon fluxes and stocks in South America and its major biomes using CMIP6 Earth System Models"

_EGUsphere, 2025_

## Author Comment (AC1)

**REPORT #2**

**General Comments:**

This manuscript assesses key components of the terrestrial carbon cycle—GPP, NPP, NEP, Ra, and Rh—over South America, with a focus on the Amazon and savanna biomes. Using output from 18 Earth System Models, the authors compare carbon fluxes across models and against literature benchmarks (e.g., CARDAMOM), and also examine differences between 'dry' and 'wet' years.

The topic is relevant and the multi-model perspective is potentially valuable, particularly for carbon cycle assessments in tropical ecosystems. However, the analysis and presentation are currently too limited and underdeveloped. Significant restructuring and clarification would be required to reach publishable quality. While I hope the comments provided are useful for future revisions, I unfortunately recommend rejection in the current form.

[Response] Thank you for taking the time to thoroughly review our manuscript and provide important feedback. We will undertake significant restructuring and clarifications in a revised version of the manuscript, to be resubmitted.

**Detailed comments:**

**Structure and framing of the manuscript:**

It's not entirely clear what the main 'results' of this paper are and often the framing changes throughout. For instance, I would suggest the novelty of this paper comes from the attempted assessment of *how well models are able to simulate carbon components in South America & its biomes, and how the models compare against one another for the region and in wet/dry years.* Yet the abstract doesn't mention the outcome of such an assessment but rather stating the patterns of the temporal evolution (which could be argued, isn't really anything new).

[Response] Thank you for your comments on how to make our work more relevant. We agree. As you noted, our focus is indeed the South America and its major biomes. The topics you have mentioned will be addressed in a revised version of the manuscript, to be resubmitted, is special in the Introduction and Abstract.

On a similar note, the introduction lacks a **justification for the research presented**, what are the research gaps and why did the authors feel that this work should be done? What is the novelty? The paragraph starting at L101 under Earth System Models is trying to argue this I think, but this should be earlier on in the manuscript and argumentation improved. The several lines on the 2024 burning seems somewhat unnecessary, more context should be added (eg. This shows vulnerability and fast changes that can occur in terrestrial ecosystems and their carbon cycles etc) or **it should be removed.**

[Response] Thank you for your comments and suggestions. We will enhance the justifications for our research earlier in the text, in the Introduction, and add more context to the text mentioning the 2024 fires, which were mentioned to provide a recent example on the importance and potential impacts of the carbon fluxes from the study region on atmospheric composition

globally.

Results/Discussions and Conclusions could be better organised. By the conclusion, several points have been repeated numerous times.

[Response] Thank you for your comment on this section of the manuscript. We will revise the text and remove repetitions.

**Models:**

**More information is needed on the models presented throughout this study and likely some further analysis is required on the following points:**

18 ESMs are used according to data availability. However, the models selected greatly vary in their set up and functionality, especially related to carbon dynamics and terrestrial processes, which will be influencing the results you show and discuss. For example, the EC-Earth3-Veg model used is a configuration that does not include ocean biogeochemistry and therefore does not have a fully coupled carbon model activated whilst other models such as CESM2 does have a fully coupled carbon model on. On the other hand, I believe that in the French IPSL-CM6A-LR model, ORCHIDEE does not have the dynamic vegetation scheme activated, which limits PFT responses and hence carbon dynamics. Whilst it is completely understandable to not have descriptions of each of the ESMs, Table 1 should be expanded to have more details of differences in the models or something similar added to SI. I also think there needs to be a discussion on how these differences impact the results added.

[Response] Thank you for the suggestion.

We will expand and include more information about the models in the supplementary material. We will add more details about each Earth System Model, and this text will provide further substantiation for Table 1.

More details should be added about what CARDAMOM is and how it works. Looking at late figures in the manuscript (e.g. Fig.8), it looks like CARDAMOM disagrees with other estimates in literature. As a reader I therefore have no indication about how 'good' CARAMOMs outputs are. Therefore, a couple of sentences about CARDAMOMS performance or sources for CARDAMOM evaluations would help.

[Response] Thank you very much for the suggestion. We plan to include a section providing further details on CARDAMON, as well as on the other research sources shown in the manuscript figures.

Simulation details: More details on the simulations that generated the ESM output in this study would be beneficial under the Earth System Models/Methods section. For example, what data was used to drive land surface change in the models? This would help later in the discussions of land use change impacts on NBP – do the model simulations include the land use changes discussed from L396 onwards? How?

[Response] Thank you very much for the comments. We will add more details on the simulations as you have suggested.

| **Biome definition:** |
| --- |
| How are the biomes presented in Figure 2 defined? Did the authors select the regions based on observations/literature/models? Figure 2 looks like perhaps it is from CARDAMOM - this should be added to the methods and figure caption. |
| [Response] Thank you for your observation. Both, the Amazon and the South American Savannas, were defined based on published literature (e.g. Olson et al (2001), Eva et al (2005), Cardoso et al (2017) and Stier et al. (2020)), and official geographical surveys from the Brazilian Institute of Geography and Statistics, cited in the text. We will revise and improve the explanation on the biome regions definition in section 2.3, "Study area". |
| The next related question is, how are these regions implemented in the analysis. Presumably each model will have slightly **different simulated regions for the 2 biomes and also different PFTs defined within them which then influences the carbon fluxes** and stocks calculated in the results. Some spatial plots of the models either in the results or as an SI figure are needed to see how spatially different the vegetation in the models are and how well they align with Fig 2. For instance, dominant PFT in each gridcell or vegetation biomass. |
| [Response] Thank you for your observation. For the development of this research, all model outputs and study region definition were first interpolated on a common regular grid with 1°x1° lat/lon spatial resolution, aligning all datasets in space and allowing us to filter for each biome in separate. We will improve the text to add more details on how the datasets were harmonised before analysis. |

| NEP: The sentence defining NEP/NEE (L78) is worded in a way that suggests NEP is always related to forests, when perhaps the authors meant just the net flux of $CO_2$ from a given ecosystem? It should also be made clear how NEP was calculated in the analysis, i.e was it calculated by looking at the net ecosystem flux for given spatial definitions (e.g masks such as in Fig.2) or is it calculated only from gridcells that contain 'forests' in the model outputs (for instance where tree cover >60% or similar). How is NEP calculated for the whole of south America (Fig.3)? |
| --- |
| [Response] We sincerely appreciate the observations. NEP was calculated by subtracting Rh from NPP at each grid cell of the ESMs. Additional details on NEP and NEE will be provided in the manuscript. |
| L152: In your selection of Wet/Dry years, the author writes that comparing observational years with simulation meteorological conditions, the results were mostly inconsistent. However, this is what is expected with ESMs. Models have their own internal variability (natural, chaotic fluctuations in the climate system that occur even under the same external forcings) and therefore timing of specific events, such as wet and dry years is often don't align with observations. It is therefore good practice not to compare observations with specific years in most ESMs. I would change the lines 152-155 to explain this and frame your approach |

accordingly.

[Response] We greatly appreciate the comments. This explanation will be added to clarify the selection of Wet and Dry years.

**Figures: In general, the size of figures should be increased relative the legend text size. The horizontal line around zero (yellow dashed line) should be explained in the captions, changed colour or removed.**

[Response] Thank you very much for the comments. In the revised manuscript, we will enhance the size and resolution of the figures, and we will also change the color of the line near zero.

Fig 6: Where does the big range in carbon pools come from? Again, this could maybe be explained by spatial vegetation plots.

[Response] Thank you for the comment. In addition to the differences in areal extent of vegetated land, the lines in Fig. 6 may also reflect differences in vegetation types and treatment of disturbances, resulting in more/less biomass and carbon stocks. As you have suggested, vegetation maps can help understand these substantial range in the models results. We will discuss this point in a new version of the manuscript, and perhaps add supplemental maps of vegetation types.

Wet/Dry years: Why did the authors choose not to do the wet/dry years analysis also for the whole of South America as they did with the temporal fluxes?

[Response] Thank you for the question. We plan to incorporate this analysis in the revised version of the manuscript.

Section 4.3: One could argue that the model results do not match well with the reference data in all cases. For example, in Figure 10. GPP and Rh reference data for the Amazonian do not match with the ensemble of models, yet the authors indicate otherwise. Perhaps a better explanation is needed here.

[Response] Thank you for your comment. We agree. Currently, the text do not provide details on the agreement between models and observations for all variables, and need improvements. In the revised version of the manuscript, we will address your observation.

**Technical comments:**

L45 lacks a comma after factors. I would also avoid the use of 'probably' here.

[Response] Thank you for the suggestion, we will correct it.

L393: EC-Earth3-Veg is listed as a model whose coupled surface model is CLM which is not the

| |
|---|
| case. |
| [Response] Thank you for the comment. We agree, and will fix that in the manuscript revision. |

---

## Author Comment (AC2)

**REPORT #1**

**Comments:**

The author provides a brief comparison of 18 Earth System Models (ESMs) from CMIP6 in simulating carbon cycle variables over South America, including global Gross Primary Productivity (GPP), Net Primary Productivity (NPP), Net Ecosystem Productivity (NEP), Net Biome Productivity (NBP), as well as autotrophic (Ra) and heterotrophic respiration (Rh). The study highlights both the consistencies and discrepancies among the model simulations. However, this work cannot be considered a proper assessment or evaluation. Firstly, it lacks validation against observational data. Secondly, it does not employ rigorous statistical analyses to support the inter-model comparisons. For example, the use of a Taylor diagram would be an appropriate and widely accepted approach for evaluating model performance. Given these shortcomings, the manuscript is not currently suitable for publication. I recommend rejecting the submission.

[Response] Thank you for taking the time to thoroughly review our manuscript and provide constructive feedback. We plan to implement additional statistical analyses to help interpret our results, and Taylor diagrams will be considered in a revised version of the manuscript, to be re-submitted.

**Detailed comments:**

Line 15: Please also show the fraction of global autotrophic (Ra) and heterotrophic (Rh) respiration.

[Response] Thank you for your observation. The sentence in line 15 now reads:

"Results show that South America accounts for 25-30% of the global Gross Primary Productivity (GPP), 21-28% of the global Net Primary Productivity (NPP), 17-50% of the global Net Ecosystem Productivity (NEP), 15-30% of the global Net Biome Productivity (NBP), 18-30% of the global autotrophic (Ra) and 44% of the heterotrophic (Rh) respiration."

Line 25: "Carbon is a critical element in the Earth system.", please give the reference.

[Response] Thank you for the suggestion. We'll include the scientific article reference.

SUAREZ, C. A.; EDMONDS, M.; JONES, A. P. Earth catastrophes and their impact on the carbon cycle. Elements, v. 15, n. 5, p. 301–306, 2019.

Line 26-28: This sentence also need corresponding reference.

[Response] Thank you for the observation. We will include this scientific article reference:

JARIWALA, D.; SRIVASTAVA, A.; AJAYAN, P. M. Graphene synthesis and band gap opening. Journal of Nanoscience and Nanotechnology, v. 11, n. 8, p. 6621–6641, 2011.

| |
|---|
| Line 35: Suggest to introduce both biogeophysical and biogeochemical paths, such as changing the energy balance on land surface. |
| [Response] Thanks for your contribution. We have changed the sentence to consider the paths mentioned:

"This increase in atmospheric CO2 has affected the global climate in several ways by influencing multiple processes in the biosphere, such as photosynthesis and plant respiration, which are influential biogeophysical and biogeochemical paths (Jung et al, 2017)." |
| Line 43: It's typo? Was it should be 0.076 PgC? |
| [Response] Thanks for pointing this out. We fixed that in the sentence:

"In 2024, extensive and persistent areas of burning in the Brazilian Amazon, Cerrado and Pantanal biomes in Brazil generated a continuous source of carbon emissions with a record that broke the record of the last 22 years of monitoring carried out by CAMS (acronym in English for Copernicus Atmosphere Monitoring Service) - approximately 0.076 PgC, 47% of which was in the month of September alone." |
| Line 60-64: Please provide more information about the model experiments, especially the MIPs (maybe DECK, historical and CORDEX) that related to this study. |
| [Response] Thank you for the suggestion. We have added more information and a reference:

"The most recent phase, CMIP6 (Eyring et al., 2016a), provides a large set of model simulations and includes 23 CMIP6-Endorsed Model Intercomparison Projects (MIPs) which facilitate a better analysis of specific scientific questions. This includes core initiatives like the DECK (Diagnostic, Evaluation and Characterization of Klima) experiments, which establish a fundamental baseline for documenting model characteristics and ensuring comparability across CMIP phases (Eyring et al., 2016a). Additionally, projects like CORDEX (Coordinated Regional Climate Downscaling Experiment) focus on enhancing climate projections at the regional scale through the intercomparison and evaluation of regional climate models (Giorgi et al., 2009)."

Giorgi, F.; Jones, C.; Asrar, G. Addressing climate information needs at the regional level: the CORDEX framework. WMO Bulletin, v. 58, p. 175–183, 2009. |
| Line 65: "discuss", it will be better to use evaluate or assess. |
| [Response] Thanks for your suggestion. We have revised the expression "discuss"

We evaluate the quality with which the CMIP6 models represent the components of the carbon balance compared to previously published and estimated results. |

Line 72: Suggesting to put figure 1 into supplementary materials.

[Response] Thank you for the suggestion. Implemented.

Line 73: "120 PgC yr-1", please give the reference.

[Response] Thank you for the suggestion. Reference included:

"The scope of fluxes and stocks in the terrestrial carbon cycle can be seen in Fig. 1. Plant photosynthesis causes a net absorption of CO2 from the atmosphere by terrestrial ecosystems of about 127 PgC yr−1 (Zhang et al, 2017)"

Zhang, Y., Xiao, X., Wu, X., Zhou, S., Zhang, G., Qin, Y., & Dong, J.: A global moderate resolution dataset of gross primary production of vegetation for 2000–2016. Scientific Data, 4, 170165, https://doi.org/10.1002/joc.4847 , 2017.

Line 82: The authors might indicated the carbon loss by hydrological process such as leaching, "water fluxes" makes people misunderstanding.

[Response] Thank you for your observation. Implemented.

"Net Biome Production (NBP) is the flux that measures the change in carbon stocks after accounting for losses due to natural or anthropogenic disturbances such as fire, deforestation, leaching, severe downbursts, and timber products."

Line 84: It was used as "cVeg" instead of "cVEg".

[Response] Thank you for your observation. Implemented.

"Vegetation carbon (cVeg) or biomass represents the carbon reservoir in the leaf, stem, root, as well as other plant components such as fruits. Carbon stored in the soil (cSoil) is found in organic (plant and animal residues, microbes and microbial by-products) and inorganic (carbon minerals produced by the weathering of the original material, or reaction of soil minerals with atmospheric carbon dioxide) forms."

Figure 1: Before explain detail of this figure, please give a general figure description.

[Response] Thank you for your observation. We have added a description.

"Figure 1. Components of fluxes and stocks in the terrestrial carbon cycle. The green flux represents carbon uptake by vegetation. The red fluxes indicate carbon release through plant respiration (Ra), soil respiration (Rh), and disturbances, whether natural or anthropogenic, such as fire. The sum of respiratory fluxes constitutes the total ecosystem respiration (Reco). The balance between gross primary production (GPP) and plant respiration (Ra) defines the net primary production (NPP). The difference between NPP and Rh describes the net ecosystem production (NEP). Finally, the balance between NEP and all carbon loss fluxes determines the net biome production (NBP)."

| |
|---|
| Table 1: Please move this table to supplementary materials, for there are too many figures and tables in the manuscript. |
| [Response] Thank you for the suggestion. The figure was moved to the supplementary material. |
| Line 101-108: This paragraph seems like discussion, please move it to discussion or just delete it. |
| [Response] Thank you for the suggestion. We'll delete this paragraph. |
| Line 134: "km2", please use superscript. |
| [Response] Thank you for your observation. We'll put in superscript and will review all other instances in the document.

"The Amazon biome (Fig. 2) has an estimated area of ~6.38 million km$^2$, with relatively low climatic seasonality,..." |
| Line 138-140: This sentence is hard to understand. |
| [Response] Thank you for your observation. We agree, and made the sentence easier to read.

"The savannas of South America (Fig. 2) have an estimated area of ~3.18 million km2, comprising a large part of central Brazil, and parts of the Brazilian states of Amapá, Roraima and Amazonas. Outside Brazil, they are also found in Venezuela, Colombia, southern Bolivia, northern Paraguay and Argentina." |
| Figure 2: The author did not clearly specify in the manuscript how the Amazonian biome area and the South American Savannas biome areas were defined, which may cause confusion for readers. It is recommended to add a corresponding legend in Figure 2 to clarify the biome boundaries and to improve the image resolution to enhance readability. |
| [Response] Thank you for your observation. We agree and have improved the description of these areas.

"Figure 2: The Amazon Biome (blue area) has an estimated area of ~6.38 million km2, with its geographical boundaries established as proposed by Eva et al. (2005), Castanho et al. (2013), Cardoso et al. (2017), Flores et al. (2024), and the IBGE (2023). The South American Savannas (brown area) cover an estimated area of ~3.18 million km2, with their geographical boundaries based on Maria et al. (2002), Schmidt et al. (2009), Bridgewater et al. (2004), and the IBGE (2023)."

IBGE. Banco de Dados e Informações Ambientais. https://bdiaweb.ibge.gov.br/#/home, last access 23 Sep 2024, 2023. |
| Line 153: Where is the results? If the authors mentioned some result, please show it in main text or supplementary materials. |

[Response] Thank you for your observation. We have' included graphical information (in the table below) showing all the dry and wet years for each model, as well as observational results proposed by Espinoza et al. (2019). This will be included in the supplementary materials.

TABLE: Determination of dry and wet years in the Amazon biome for each ESM, based on the methodology proposed in this study. Orange blocks represent dry years, blue blocks represent wet years. The first column corresponds to the classification proposed by Espinoza et al. (2019), Table 1.

Espinoza, J. C.; Ronchail, J.; Marengo, J. A.; Segura, H. Contrasting North–South changes in Amazon wet-day and dry-day frequency and related atmospheric features (1981–2017). Climate Dynamics, v. 52, n. 9–10, p. 5413–5430, 2019.

Line 165: Is this result supported by any statistical analysis?

[Response] Thank you for the question. In order to evaluate if the trends we found were significant, we have applied the Mann-Kendall tau coefficient analysis, following Mann (1945) and Kendall (1975).

Mann, H. B. Nonparametric Tests Against Trend. Econometrica, v. 13, n. 3, p. 245, jul. 1945.

Kendall, K. Thin-film peeling-the elastic term. Journal of Physics D: Applied Physics, v. 8, n. 13, p. 1449–1452, 11 set. 1975.

Line 173: Also need statistical evidence.

[Response] Thank you for the observation. We agree. We will improve this and several other parts of the manuscript to provide statistical evidence to our findings.

Figure 3: what is the yellow dash line and horizontal line represent respectively?

[Response] Thank you for the observation. The dashed yellow line represents the zero line, which we believe it is interesting to highlight. The figure caption will include information on that.